# Semantic Aware Just Noticeable Differences for VVC compressed Text Screen Content Images

## ABSTRACT

With the rapid development of multimedia applications such as online education, remote conferences, and telemedicine, an emerging type of image known as text screen content images (TSCI) has gained widespread utilization. Compared to natural images captured by cameras, TSCI is generally generated or rendered by computers and exhibits significant differences in content characteristics. One of the significant differences is that TSCI mainly contains text, which is a symbol system defined by humans with specific semantics. As an important carrier for transmitting semantic information, the quality of text in TSCI significantly affects the subjective perception experience of multimedia system users. Just noticeable difference (JND) is a widely studied image quality measure that is theoretically closest to human perception. However, the traditional JND (T-JND) tests fail to distinguish text from other image contents, ignoring the significant impact of semantic readability of text on image quality. This paper focuses for the first time on the impact of text semantics on the quality of TSCI, and a semantic aware JND model for TSCI compressed by the state-of-the-art versatile video coding (VVC) standard is explored and discussed. Specifically, a matching TSCI database is first established. Using the database, image subjective observation comparison experiments are further designed and carried out to construct the traditional JND (T-JND) as well as the semantic aware JND (S-JND). By comparing the experimental results, crucial conclusions are reached, including the fact that the S-JND provides a more precise description of the quality of TSCI compared to the T-JND. These conclusions have important guiding significance for the subsequent development of efficient JND models suitable for TSCI compressed by VVC.

## CCS CONCEPTS

• **Human-centered computing** → **User models**.

## KEYWORDS

Just noticeable difference, Screen content image, Text, Quality assessment, Quality of experience

## 1 INTRODUCTION

Screen Content Images (SCIs) which typically feature a combination of text, graphics, tables, and potentially natural contents, have become a ubiquitous presence in various screen based multimedia platforms and devices for supporting online education, screen-sharing, cloud computing systems, remote conference, product advertising [6, 8, 19, 24], etc. Text, as an important visual content in SCI, comprises the primary content of the Text Screen Content Image (TSCI). Text is composed of a series of characters which are important carriers of human knowledge transmission. Text is composed of a series of characters which are important carriers of human knowledge transmission. Generally, then content in TSCI is semantic sensitive and is closely related to human prior knowledge. It is reported that human eyes are is highly sensitive to the quality of text [35].However, due to limited bandwidth and storage ability, TSCIs are generally compressed before transmission, which introduces inevitable image quality degradation. Thus, the quality of TSCI plays a pivotal role in enhancing the informational value, online interactivity, and accessibility of digital media.

Compared with the camera captured natural images, TSCI consists of sharp edges, thin lines with few colors, even one-pixel-wide single-color lines [16]. Also TSCI presents two main properties: limited number of colors and repeating patterns [24]. To improve coding efficiency of text information, various coding tools are introduced in HEVC Screen Content Coding extensions (HEVC-SCC) and the state-of-the-art Versatile Video Coding (VVC) standard [32, 38], such as the Intra Block Copy (IBC), adaptive color transform (ACT), palette mode, Transform Skip Residual Coding (TSRC), and Block-Based Differential Pulse-Code Modulation (BDPM), etc. However, when SCIs are processed, various distortions are still inevitably involved [29]. Considering human visual system (HVS) is the ultimate receiver and appreciator for the majority of processed images, video and graphics [16], accurately assessment of the perceptual quality of human observers is of great significance.

The ITU-R Recommendation BT.500 [5] is one of the most widely used subjective image quality assessment protocols, which defines standardized testing procedures for evaluating the perceptual image quality based on human observer judgments. And the Mean Opinion Score (MOS) or the Differential Mean Opinion Score (DMOS) methods for subjective scoring are introduced [38]. Previous studies have suggested that human perception and judgment in psychophysical measurement tasks typically perform better in comparison tasks than in absolute ratings [32]. Therefore, by compare between two visual stimuli, Just Noticeable Difference (JND) which is defined as the smallest difference as discernible by human, has been widely used for predict image coding distortion levels [31]. JND is a psychophysical concept, and according to Weber's Law, also known as the Weber-Fechner law, the noticeable difference in a stimulus is proportional to the intensity of the stimulus [31]. In the area of quality assessment, JND focuses on capturing the tipping points of changes in image/video quality. As such, JND represents a widely studied measure of image quality that closely aligns with human perception, offering a theoretical framework that resonates deeply with our subjective experiences.

In the past years, researches on JND in the video and image fields include obtaining subjective JND data, constructing databases, or using JND models to validate and improve objective quality assessment algorithms for multimedia signals [37]. To solve the problem of insufficient databases, several image/video quality assessment datasets based on JND have been developed, and subjective experiments have been conducted to search for JND points [9, 12, 15, 23, 27]. In fact, human eyes cannot sense any changes below the JND threshold due to their underlying spatial-temporal masking properties [4]. Based on the above description, more and

more JND models [3, 14, 25, 29, 33] have been proposed to optimize the video coding and compression process by determining the smallest difference that can be perceived by the human eye in terms of visual perception, thus improving the coding efficiency and subjective visual experience [36].

However, the current JND datasets still treat the texture in TSCI as the same as that in natural images. Actually, humans read the text, thus JND thresholds should be semantically aware, which means that quality changes for TSCIs should consider the quantity and accuracy of semantic information transmission. Without considering its readability, traditional JND based datasets are not able be directly applied into TSCI related applications. And, this has not been effectively considered in the development of traditional subjective JND experimental design.

An important prerequisite for establishing an effective JND model is to design reasonable subjective observation experiments to obtain accurate JND observation data from subjects. To achieve this goal, the accurate definition of the "difference" is a key process for determining accurate JND observation data. In traditional JND observation experiments, the definition of the "difference" is vague and only texture change has been considered, which is not suitable for TSCIs with semantic information. When viewing TSCIs, the degree to which the human eye can effectively extract semantic information from the text is a crucial factor in determining the perceived quality of TSCIs. Therefore, it is necessary to focus on the influence of text semantics on the definition of difference when constructing subjective observation experiments for TSCI.

In this paper, focusing for the first time on the impact of text semantics on the quality of TSCI, two different subjective JND experiments are introduced, one of which is the same as the Traditional JND experiment (T-JND), and the other is the Semantic Aware JND experiment (S-JND). Based on a new established TSCI database, denoted as Semantic Aware Just Noticeable Differences based TSCI (SAJ-TSCI) database, the number and the location of the JND points are compared. And then the cognitive processes are analyzed and come into crucial conclusions based on the observation time and the eye movement data. The rest of the paper is organized as follows: Section II covers related work. Section III outlines the construction of the dataset and the design of the subjective experiment. Section IV conducts a statistical analysis of the experimental data. The results of the data analysis are discussed in Section V. Finally, conclusions are drawn in Section VI.

## 2 RELATED WORK

The main purpose of Image Quality Assessment (IQA) is to propose an assessment method that can accurately measure the subjective perception of the human eye based on the perceptual characteristics of the Human Visual System (HVS). Subjective assessment methods yield results that are closer to human perception than objective assessment methods, e.g., PSNR, MSE.

The concept of JND is widely applied subjective assessment method, and it determines how accurate human sense are [37]. [30] proposes a way to use JND for video quality measurement which adopts "pair comparison" or "two-alternative forced-choice", where subjects are asked to determine which of the two videos, i.e., the

original source and the compressed one, is more distorted. However, the testing time required for each subject in this method is very long, which may lead to fatigue or other factors that affect the experimental results. In [15], the first JND dataset is constructed, consisting of 5 JPEG encoded images and 5 H.264/AVC encoded videos. Combining with the method in [30] with a binary search method, the proposed observing method further reduces the size of subjective JND experiments, and K-means clustering is used to process the experimental data. In [9] the Gaussian Mixture Model (GMM) method is used to model the JND observing data, and by comparing GMM with K-means methods, it is proved that the GMM method provides more accurate quality levels and is actually more reasonable. To solve the problem of insufficient databases, several image/video quality assessment datasets based on JND are developed. The MCL-JCI dataset [12] contains 50 source images, each having a resolution of 1920×1080, each source image is encoded 100 times with the JPEG encoder. Subsequently, by analyzing and post-processing the original JND data, the staircase quality function (SQF) is accessed. A video quality dataset MCL-JCV [27] encoded with H.264/AVC is constructed, with quantization parameter (QP) values ranging from 1 to 51. JND points of 50 subjects are recorded relative to each video segment. The difference between every two adjacent JND points is calculated, and an outlier detection algorithm is proposed to remove unreliable data. [23] establishes a JND dataset for VVC standard which consists of 202 images with a resolution of 1920×1080. Each image is encoded by VTM 5.0 intra coding with the QP ranging from 13 to 51. In addition, there are also some JND experiments designed for specific problems and application programs. [18] studied three charts: bar charts, pie charts, and bubble charts, analyzing the relationship between JNDs and two main visual variables: the intensity of visual elements and their distance.

By combining the features of JND and HVS, such as contrast sensitivity, brightness masking, etc., some JND models have been formed in image/video perceptual compression. JND-based models aim to find the maximum distortion levels that cannot be perceived by the HVS, and use them to eliminate the maximum tolerable perceptual redundancies [21]. The JND model has been established in [34] and [22] by using luminance masking effect and contrast masking effect. The JND model in [3] and [2] introducd Contrast Sensitivity Function (CSF) into the model to improve the accuracy of assessment. In addition to the aforementioned JND models, other JND models are combined with machine-learning techniques [14, 17, 39].By removing the perceptual redundancy information according to the JND levels, compression gain can be further realized [36]. JND models have been used in image compression to improve the overall compression efficiency [20], guide the quantization process in coding [7, 26], and achieve more efficient perceptual encoding.

By combining the features of JND and HVS, such as contrast sensitivity, brightness masking, etc., some JND models have been formed in image/video perceptual compression. The JND model has been established in [21, 25]by using luminance masking effect and contrast masking effect. The JND model in [2, 3] introduces Contrast Sensitivity Function (CSF) into the model to improve the accuracy of assessment. In addition to the aforementioned JND models, other JND models are combined with machine-learning techniques [14, 17, 39]. By removing the perceptual redundancy

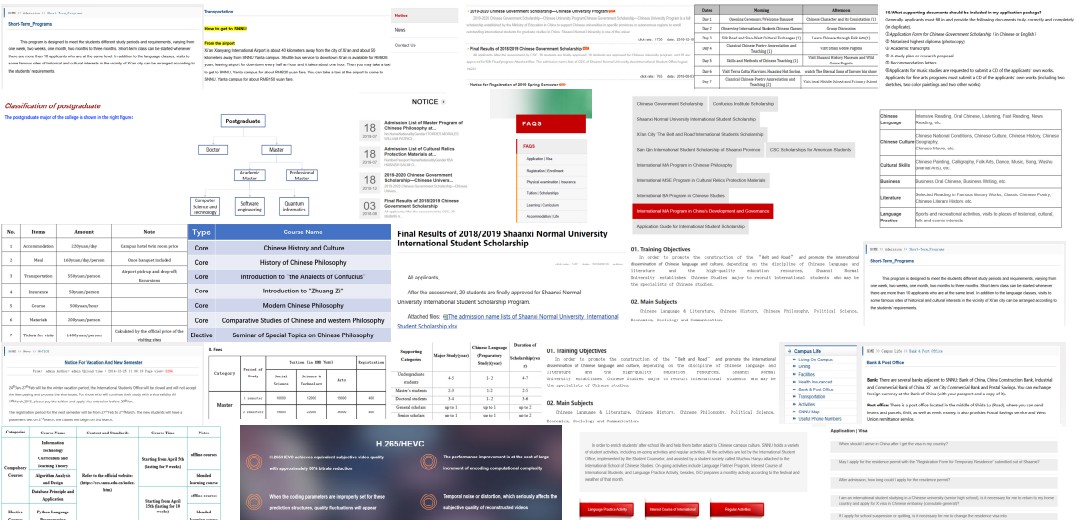

**Figure 1: Representative thumbnail images of 24 source sequences in the dataset.**

information according to the JND levels, compression gain can be further realized [36]. Currently, JND models have been used in image compression to improve the overall compression efficiency [20], guide the quantization process in coding [7, 26], and achieve more efficient perceptual encoding.

Although JND based quality assessment methods for natural images have been widely studied and developed, the current methods also exhibit disadvantages:

- The current SCI databases or JND based databases are developed for natural images, and as far as we know there has no dataset been proposed for TSCI.
- Compared with the natural images, the textual of the text in TSCI contains a lot of key information, and human eyes are sensitive to the semantics of the textual information [30]. However, the traditional JND (T-JND) tests fail to distinguish text from other image contents, ignoring the significant impact of semantic readability of text on image quality.

## 3 SUBJECTIVE EXPERIMENTS

### 3.1 Test Material: SAJ-TSCI Dataset

The current JND dataset includes MCL-JCI [12], MCL-JCV [27], VideoSet [28], etc., developed for the coding standards including JPEG, H.264/AVC, VVC, etc. But these datasets have the following obvious problems: 1) Most of these databases are developed for natural images, and the TSCI still remain untouched; 2) Comparing between the reference image and the distorted images, the definition of the "difference" is not clear, and semantic information in text is not involved. Therefore, in order to facilitate the study of the differences between T-JND and S-JND for the TSCIs, a text screen content image dataset, denoted as SAJ-TSCI, is established firstly. The dataset contains 24 original TSCIs, as shown in Figure 1. All of the TSCIs in this dataset are from web pages. The spatial resolution range of the original image is 320 to 1210 pixels, the color space of the image is YCbCr 4:4:4, and the bit depth is 8.

According to the ITU-R BT.1788, the spatial perceptual information (SI) is used to measure the spatial detail information of the original images [10]. In order to adapt to the sharp edges and even single pixel boundaries of TSCI texture, this paper uses the SI to show the spatial features of the 24 original images in the SAJ-TSCI dataset as shown in Figure 2.

The Weber-Fechner law [31] shows that human eyes cannot perceive a small quality difference between two images, the human eye can perceive quality differences only when the quality difference exceeds a certain threshold. When the image is encoded with a smaller QP, the difference between the reconstructed image and the original image is smaller, which is difficult to be perceived by the human eye. When the image is encoded with a larger QP, the reconstructed image is totally blurred, and is difficult to extract any useful information. Thus, the QPs considered in this paper range from 28 to 58, and VTM16.2 is was used to encode the original image to obtain the reconstructed images corresponding to each QP. And 744 reconstructed images compressed by using the VVC standard can be obtained.

The JND subjective tests correspond to the proposed dataset is constructed to answer the following questions:

- Is the number of T-JND points that humans can distinguish from S-JND points the same?
- If different, which quantity is larger? What are the differences in their boundary points?
- What factors may have led to the difference between the two kinds of JND points?

### 3.2 JND Subjective Observation Experiment for TSCI

In order to avoid the human eye obtaining all semantic information of the text by viewing clear images at the start of the experiment, which leads to the human eyes having prior knowledge of text for viewing subsequent distorted images. The initial image is set as the image related to largest QP, i.e., the images with the largest

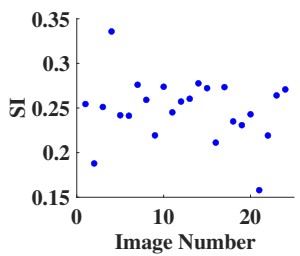

**Figure 2: SI information of the original images in SAJ-TSCI dataset.**

distortion without any semantic information. As the image becomes clearer, the amount of the semantic information that subjects can obtain gradually increases. Within each JND level, the reference image remains static in the test while the target varies. For the beginning of the test, the reference image is the image of QP=58. The targets are arranged successively in accordance with decreasing QP. If a target is deemed to be visually indistinguishable from the reference by an observer, then the next target is presented. This continues until the target list has been exhausted or when the target becomes perceptibly different from the reference. If the two compared image is visual different according the judgment questions, the target is marked as a JND threshold.

### 3.3 Definition of "Difference" in the Experiment

The purpose of the T-JND experiment is to determine what extent of the image distortion or changes can be detected by the observer. Currently, in T-JND experiments, currently the definition of the "difference" is not uniform and clear among different methods. This may lead to a lack of clear guidance or understanding for subjects to perform tasks. Most of the subjects may make choices based on their own understanding of notifiable " differences". However, TSCI is semantic aware, in the S-JND experiments, subjects needs to pay attention to the difference in the amount of the semantic information that can be obtained from the images. And the assessment questions for the T-JND and the S-JND in this paper are set as:

**Questions for the T-JND:** Is there any difference between the two images displayed on the screen?

**Questions for the S-JND:** Is there any difference in the amount of semantic information you can obtain from the two images displayed on the screen?

### 3.4 Experimental Setup

49 subjects, including 23 males and 26 females, aged between 20 and 25 years old took participation in the experiments. All subjects have no research experience in the field of image quality assessment and have normal vision. All subjects recruited for this experiment were graduate students. Considering that English semantic information in TSCI needs the subjects to recognized in this experiment, all subjects were required to pass the national English proficiency test. Before the start of the testing step, each subject received simple training and a PPT introduction document to introduce them to the usage of the graphical user interface, as well as the testing steps of this experiment.

Considering the visual fatigue that occurred during the experiment, T-JND and S-JND tests were conducted separately. The subjects were first introduced to the T-JND experiment, and after two weeks, the S-JND experiment was introduced again. In order to further alleviate the pressure on the subjects, all 24 sets of image sequences were divided into 4 parts, each containing 6 original images. Each subject needs to complete the viewing of 2 parts of image sequences to ensure that each set of image sequences is observed by 30 subjects. Each subject was required to complete a comparison of 1 part of image sequences at once, and to rest for at least 20 minutes before continuing to compare the remaining parts. The experiment used a 2K monitor with high resolution, wide color gamut, and accurate color representation capabilities to ensure a stable and consistent visual assessment environment for the subjects. The background of the graphical user interface was set to pure gray, and all prompt information was displayed in black font. The lighting conditions in the laboratory were normal. The experimental setup and process comply with ITU-R BT.500-13 standards [5].

Considering semantic acquisition, if two images are presented on the screen simultaneously, the amount of information the subject obtains from both images may affect each other. Therefore, in this experiment, the Double Stimulus (EBU) method was cyclic in that the assessor was first presented with an anchor image, and then with the compare image. Meanwhile, there was no time limit for either experiment. Subjects clicked "Yes" or "No" according to their viewing situation by comparing the two presented images. The personal information of the subjects, QP for each JND level, and their action time during the experiment have been recorded for subsequent analysis. In addition, Tobii EyeX was used in the S-JND experiment to record eye movement data for subsequent analysis.

### 3.5 Outlier Detection and Normality Verification

For the all received JND testing results, outliers are firstly detected and rejected according to the method in [23], which detects the consistency between a certain subject and the original results of all other subjects during the testing period.

The subjective data of each image is processed separately. Detect outliers by checking the consistency of the original results between a specific subject and all other subjects during the testing period. Firstly, standardize the raw data of each image to calculate the z-score.The average value $\mu_n$ and standard deviation $\sigma_n$ of the original samples in the $n^{th}$ image set are calculated as (1) and (2), respectively,

$$\mu_n = \frac{1}{M} \sum_{m=1}^{M} Q_n^m \tag{1}$$

$$\sigma_n = \sqrt{\frac{1}{M-1} \sum_{m=1}^{M} (Q_n^m - \mu_n)^2} \tag{2}$$

where $Q_n^m$ represents the collected samples of the $m^{th}$ subject on the $n^{th}$ image set, where $m \in 1, 2..., M$, $n \in 1, 2..., N$.

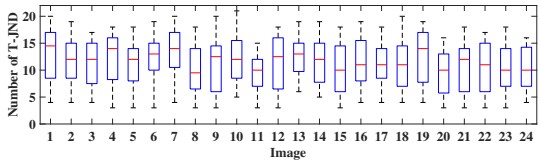

(a) The number of JND points in the T-JND experiment

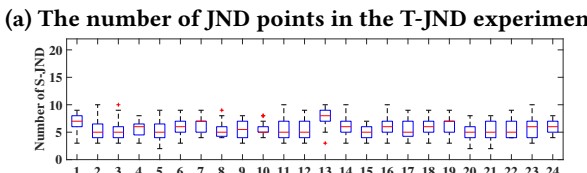

(b) The number of JND points in the S-JND experiment

Figure 3: JND points quantity box plots.

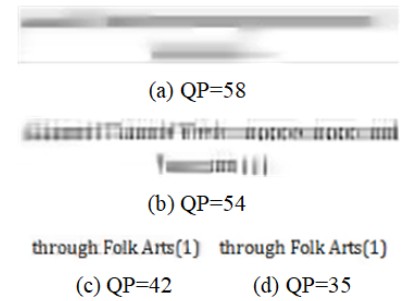

(a) QP=58

(b) QP=54

through Folk Arts(1)    through Folk Arts(1)

(c) QP=42          (d) QP=35

Figure 4: Examples of distorted images.

Calculate the z-score using (3):

$$Z_n^m = \frac{Q_n^m - \mu_n}{\sigma_n} \tag{3}$$

Then, the shape of the distribution is evaluated by calculating the kurtosis of the original data. If the distribution follows a normal distribution, the score is considered reliable. Next, for each JND points of each image, check the relationship between the JND point of that image and the mean and standard deviation of the rating set. If the JND point exceeds a certain threshold range, it will be marked as an outlier and discarded. The details of these steps are summarized in [22].

After removing samples from outliers, the distribution of T-JND and S-JND data go through normality testing by using Jarque-Bera test [11] as shown in (4). After removing outliers, the remaining T-JND points follows a normal distribution along with the S-JND points.

$$JB = \frac{n}{6}(s^2 + \frac{(k-3)^2}{4}) \tag{4}$$

where $n$ is the sample size, and in this paper, $n$ is the raw JND points for a certain test image, $s$ is the sample skewness and $k$ is the sample kurtosis.

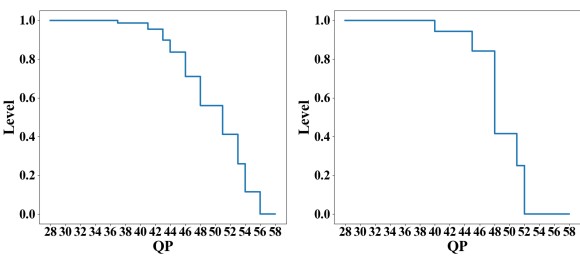

Figure 5: The box plots of the lowest JND locations. The top figure shows the results of the T-JND experiment, and the bottom figure shows the results of the S-JND experiment.

## 4 ANALYSIS AND DISCUSSION OF SUBJECTIVE EXPERIMENTAL RESULTS

### 4.1 Comparison of the Number of JND Points

The T-JND and S-JND points related to the values of QP were collected for each subject. First, the total number of JND points is analyzed for each test images in the dataset, which in turn determines the number of perceived quality levels. As shown in Figure 3, the bottom and the top of each box indicate the 25th and 75th percentiles of the samples, respectively, and the middle line is the average number of the perceived JND points.

It is found that the distribution range of the number of the T-JND points is relatively wide. As shown in Figure 3 (a), for the total number of the JND points that can be perceived by the subjects, the maximum number can reach up to 21, while some subjects can only perceive 3 JND points. This is related to the understanding of "difference" among various subjects in the first test for T-JND. Since no strict definition of the "difference" was provided to the subjects, they make their decision according to their own understanding. Based on the results in Figure 3(a), the subjects can be divided into two categories: texture perceivers and semantic perceivers.

Texture perceivers attempt to compare texture differences between two images, and as long as they feel a slight change in texture, they recorded that there is a "difference". For example, in the two images in Figure 4 (a) and (b), the subject cannot perceive any semantics information, but the texture of in the image has changed, which resulted in differences for texture perceivers. Similarly, when all the semantic information in the image can be obtained as shown in Figure 4 (c) and (d), texture perceivers still perceive differences within the stokes of characters, even if they can no longer obtain additional semantic information at this time. Therefore, due to the sensitivity of the human eye to the text region, texture perceivers collect the highest number of JND points.

For the semantic perceivers, although we did not remind them to pay attention to the semantic information, they are more inclined towards image utility. They determine a JND point when there is a difference for semantic information. For example, in a sentence, if only a few letters can be seen clearly, but the meaning of the entire sentence is still not able be acquired accurately, then the semantic perceiver believes that the difference does not exist. This leads to fewer JND points obtained by semantic perceivers.

But in the S-JND experiment, the "difference" was described and reminded to all subjects to focus on the semantic perception. It

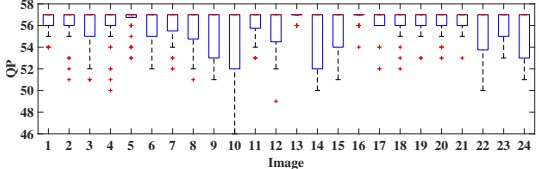

(a) The highest JND locations in the T-JND experiment

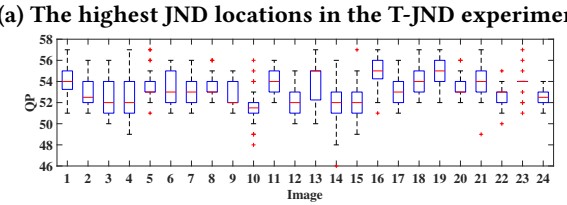

(b) The highest JND locations in the S-JND experiment

Figure 6: The box plots of the highest JND position for No. 15 image.

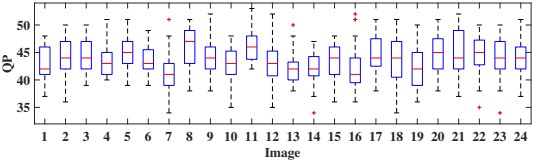

(a) The lowest JND locations in the T-JND experiment

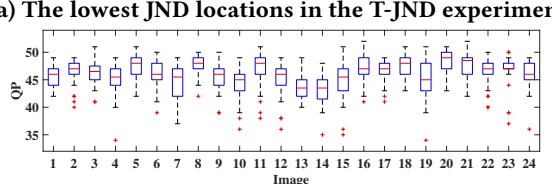

(b) The lowest JND locations in the S-JND experiment

Figure 7: The box plots of the lowest JND position for No. 15 image.

can be seen from Figure 3 (b) that the distribution range is more concentrate for S-JND, and there is no significant difference in the number of the JND points among subjects. It can be seen that the maximum number of JND points in the S-JND experiment is 10.

For the same image, different subjects still have different numbers of JND points. Therefore, GMM in [9] was used to achieve the stair quality function (SQF) for each images. Assuming the JND distribution is in the GMM form of $N$ components. The distribution of the quantified parameter in mathematics can be expressed as (5):

$$f(x) = \sum_{i=1}^{N} \pi_i \mathcal{N}(\mu_{x,i}, \sigma_{x,i}^2) \quad (5)$$

where $\pi_i$ is the mixture weight satisfying the constraint of $\sum_{i=1}^{N} \pi_i = 1$. Each component is a normal distribution that satisfies the mean $\mu_{x,i}$ and variance $\sigma_{x,i}^2$ calculated by (1) and (2).

The Expectation Maximization (EM) algorithm is used to iteratively update the parameters in GMM until the process converges or reaches the preset maximum number of iterations. More information can be found in [9].

After building the GMM, the next step is to build the corresponding Stair Quality Function (SQF). Approximating the JND distribution as the sum of $N$ peaks, as shown in (6),

$$JND(x) = \sum_{i=1}^{N} H_i \delta(x - \mu_{x,i}) \quad (6)$$

where $\delta(\cdot)$ is the Dirac function. $H_i$ denotes the posterior distribution of the $i^{th}$ component.

After building the GMM, the next step is to build the corresponding SQF. Approximating the JND distribution as the sum of N peaks, and SQF is obtained by integrating the JND function from the largest QP to the smallest QP.

One example to illustrate the relationship between SQF of T-JND and S-JND for No.15 image is given in Figure 5. It is seen that the total number of the quality levels obtained in the S-JND experiment

is smaller than that in the T-JND experiment. This is consistent with the previous analysis results.

## 4.2 Comparison of the Locations of JND Points

The box plots of the highest and lowest JND locations of the images in Figure 6 and Figure 7. Similar to the analysis above, The highest JND in T-JND experimenters concentrated into the value of QP=57 as shown in Figure 6 (a). This means that most of the observers only judge the difference by the texture difference since no semantic information is provided in this QP. The highest JND position in the S-JND experiment is lower than that in the T-JND experiment as shown in Figure 6 (b). Taking No. 15 image as an example, in the T-JND experiment, the range of the highest JND position is 51 to 57, while in the S-JND experiment, it is 47 to 55. The highest JND in S-JND is related to the semantic information increase which generally corresponding to a lower QP compared with the T-JND.

At the same time, when all the semantic information provided by the image is obtained even though with small texture changes, it will generally receive no more attention, which makes the lowest JND position in the S-JND experiment mostly higher than in the T-JND experiment, as shown in Figure 7 (a) and (b). Still using No. 15 image as an example, the lowest JND position ranges from 36 to 47 in the T-JND experiment whereas it ranges from 40 to 51 in the S-JND experiment.

## 4.3 Cognitive Process Analysis Based on Observation Time

In order to further analyze the cognitive process of the semantic information by the subjects, the observation time of each distorted image corresponding to each image in the dataset was recorded. Figure 7 shows the results from two subjects with the observing time of 6 test images in the dataset. As described in Section 3, the initial test distortion image is set as the largest QP, and then the QP gradually decreased to obtain all the JND points for a test image based on the subjective results. The enlarged figure shows

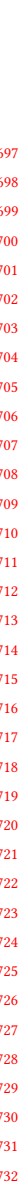

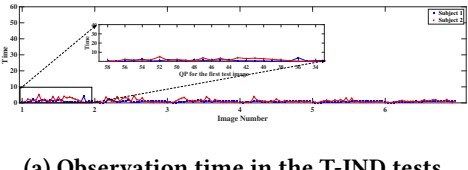

(a) Observation time in the T-JND tests

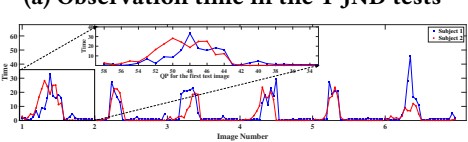

(b) Observation time in the S-JND tests

Figure 8: Observation time of Subject 1 and Subject 2 of the 6 original images with related reconstructed images in a test image group for the T-JND test and the S-JND test, respectively.

the specific observing time for each distortion images of the first test image of the two subjects.

It is seen in Figure 8 (a) that the spent time variation is stable and stays in a low level for the T-JND test. For this kind of JND test, the subjects generally pay attention to the texture differences without recognize the characters in the text. In other word, the strokes in the text are only a special complex line and the subjects did not spend more time to recognize the semantic information in the text. Thus, the observing time in the T-JND test is less since they just need to judge that whether the texture in the target image has difference compared with the reference image.

Compared to the T-JND, the subjects in the S-JND tests generally spent more time for each distortion images and appeared to exhibit a certain trend as shown in Figure 8 (b). Since the target images were arranged with a descending distortion order, the reference and the target images were started with significant distortion. At this stage, no semantic information can be perceived which makes the subject can make their determination quickly and move to the next comparison. As the QP value decreases, characters or words in the text began to be seen. At this stage, the subject needed to pay attention to increased semantic information, and the observation time increases until all information can be obtained, and the observation time reaches a peak. Afterwards, the QP value continued to decrease, however, the subject found that there would be no further additional semantic information appeared, so observing time drops again.

The above phenomena can be explained through the cognitive process of the subject. In the S-JND test, the target image starts with the largest distortion which means no semantic information can be extracted. Then, when semantic information gradually appeared, the subjects must spend more time to read/observe before they made the determination. The cognitive load theory suggests that the human cognitive system is limited [1], and processing multiple tasks or information simultaneously can lead to an increase in cognitive load. The Multimode Model proposed by Johnson and Heinz [13] suggests that attention helps to make choices about

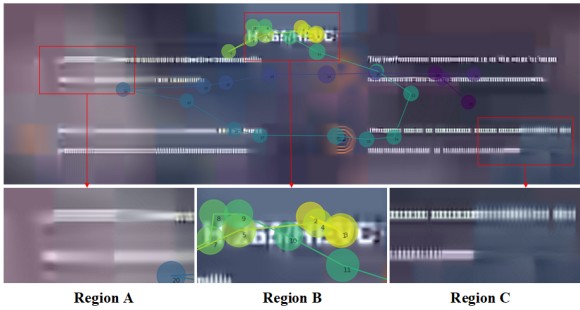

(a) QP=55

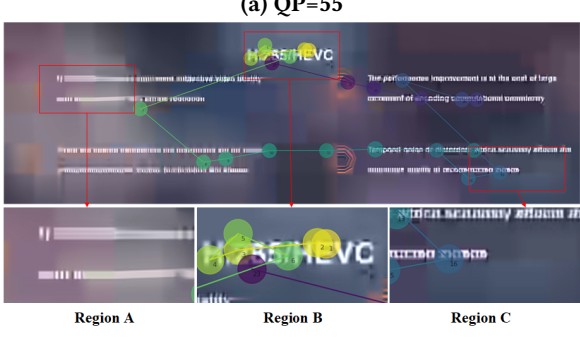

(b) QP=53

Figure 9: The eye tracking traces of Subject 1 for No. 22 image.

information at different stages of cognition. According to their perspective, attention can perform the function of selection across sensory stage, semantic stage, and conscious stage. At the sensory stage, it suggests that only physical stimuli will be processed and sensory representations can be established. And the cognition of the T-JND test mainly stays at this stage to perceive texture differences. Generally, more information processing consumes more cognitive resources. In the T-JND experiment, the subjects did not pay extra attention to the semantic features. At the semantic stage, the cognitive system constructs semantic representations of stimuli. In the S-JND experiment, semantic acquisition tasks involve understanding, interpreting, and reasoning input information, requiring deeper cognitive processing. The cognitive load of such tasks is usually high, and they require more attention and cognitive resources to process complex semantic information. This ultimately leads to the variation trend where the time spent by the subjects during the observation process, i.e., the observation time increases first and then decreases with the increase of the semantic information of the target distortion images.

## 4.4 Cognitive process analysis based on Eye Movement Data

For a further analysis of the cognitive process of the observers, eye movement data of a subject was recorded using a Tobii EyeX eye tracker in the S-JND experiment. In Figure 9, the size of the circle represents the length of the time for each eye gaze, the order of eye movements is represented by the numbers and colors. The order of

viewing gradually darkens the color of the circles from yellow to purple.

Based on the eye tracking results, the following conclusions can be draw.

- **Cognitive process is skim around when no sematic information can be obtained.**

  In Figure 9, the two images both provide no semantic information with difference in texture, however, the observation paths of the subject are similar. The eye gazes started from the center of the image, and then skimmed around and finished the observation of this image. Even though the subject has been reminded to pay attention to the semantic information, the eye traces show that the subject only roughly observed the structure of the image around and then provide a choice. This is because of no semantic information can be obtained from the two images.

- **Cognitive processes focus on the location of semantic information when clear information is the minority.**

  With the increment of the image quality, some characters were able to recognize, and as shown in Figure 10 (a). At this stage, characters that are beneficial for text semantic understanding is the saliency contents. As shown in the "Region A" of the Figure 10 (a), the blur region has no eye gaze points, while most of the eye gaze points are concentrated in the "Region C". When most of the information in a sentence is unidentifiable, subjects take the regions where he can access the semantic information as the saliency contents and spend less time on the unrecognizable text.

- **Cognitive processes focus on the blur text that disturbs the semantic information when clear information is the majority.**

  In Figure 10 (b), more eye gaze points locate in the "Region A". It is seen that most of the textual information in the first sentence from the left is recognizable, only the text in the "Region A" is blurred. At this stage, in order to finish the cognition of the complete meaning of the sentence, subjects concentrated on the blurred information, and this results in a clustering of eye gaze points in this region.

- **Cognitive processes followings the reading order when all sematic information is obtained.**

  In Figure 10 (c), all text semantic information can be obtained, and all eye tracking points are evenly distributed in the text area and follow the reading order, Generally, this stage corresponding to the lowest S-JND point.

Based on the discussion in this section, text has significant semantic features, and the loss of semantic information caused by TSCI compression can affect human cognitive processes and ultimately affect the distribution perceived quality of the TSCI. Therefore, quality evaluation methods for TSCI or effective image compression methods should be sematic aware, and accurate acquisition of text semantic information must be considered.

## 5  CONCLUSION

In this study, "difference" in the JND experiment was defined as the difference in perceived semantic information, known as the S-JND. Through subjective observation experiments, T-JND and S-JND data were collected and compared in quantity and location. The results

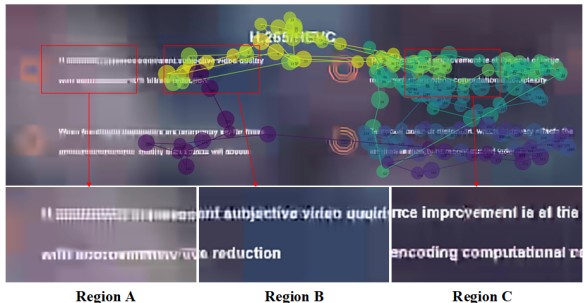

**(a) QP=51**

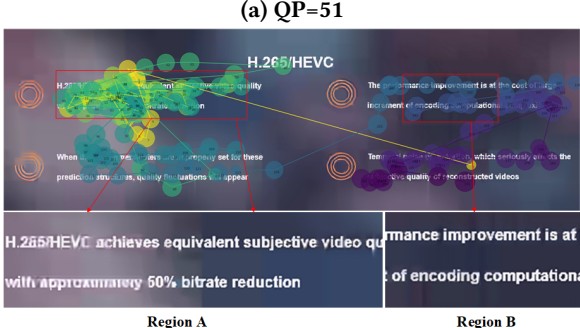

**(b) QP=47**

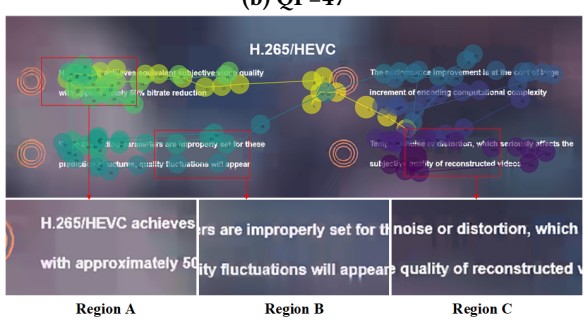

**(C) QP=44**

**Figure 10: The eye tracking map of Subject 1 is related to the reconstructed images with QP values of 51, 47 and 44 for No. 22 image.**

showed that S-JND is more suitable for measuring the quality of TSCI than the excessive quality levels in the T-JND experiment. In addition, cognitive process based on the observation time and the eye track data were analyzed. The results in this study may contribute to TSCI quality assessment and image compression for multimedia applications.

## 6  ACKNOWLEDGMENTS

This work was supported in part by the xxxxx under xxxxx.

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
