# OpenReview forum: "Semantic Aware Just Noticeable Differences for VVC compressed Text Screen Content Images"
_acmmm.org/ACMMM/2024/Conference — MM2024 Poster_

### Official Review · Reviewer_av1z · 2024-05-19

**Rating:** 3
**Confidence:** 3

**Summary:**

This paper focuses on the development of a semantic aware Just Noticeable Difference (JND) model for text screen content images (TSCI) compressed by the versatile video coding (VVC) standard.

**Strengths:**

This article has certain significance and depth in its research.

**Limitations:**

This paper focuses on the development of a semantic aware Just Noticeable Difference (JND) model for text screen content images (TSCI) compressed by the versatile video coding (VVC) standard. The paper highlights the importance of considering text semantics in image quality assessment, especially for TSCI which mainly contains text. By conducting subjective observation experiments and comparing traditional JND (T-JND) with semantic aware JND (S-JND), the study concludes that S-JND provides a more precise description of TSCI quality. The research emphasizes the need to incorporate text semantics in JND models to improve the subjective perception experience of multimedia system users.
However, there are still many areas that need improvement in this paper.

1. Describe the viewing distance during the experiment.
2. Only 24 sets of original images and 280 distorted images are too few to be representative.
3. How can this study be applied to image processing, such as image quality assessment? Please propose a specific algorithm.

**Suitability:**

2

---

### Official Review · Reviewer_8Py3 · 2024-05-23

**Rating:** 3
**Confidence:** 3

**Summary:**

This paper introduces two JND types for VVC compressed text screen content images. Specifically, a TSCI-JND database is established, in which the image subjective observation comparison experiments are further designed and carried out to construct the T-JND and S-JND. By analyzing the results in the subjective experiments, S-JND is proven to provide a more precise description of the quality of TSCI compared to the T-JND.

**Strengths:**

1. The motivation of this work is clear;
2. The established dataset is crucial for promoting research on JND and SCI images;
3. The analysis and conclusion of eye tracking in Figure 9 and Figure 10 is sufficient and confidence.

**Limitations:**

1. The organization of this paper is not well. I suggest the author should summarize this work's contributions in the introduction. Besides, it is confusing for me to read the line 163-line168. The related work should focus on the JND work, rather than the IQA.
2 There are too many mistakes in the manuscript,
 2.1 such as line 52-line 55;
 2.2 human visual system (HVS) line 80, Human Visual System (HVS) line 166
 2.3 [23] establishes a JND dataset for VVC standard which consists of 202 images with a resolution of
   1920×1080.  Such words are not very professional. I suggest "Shen et al. established a JND dataset for
    VVC standard which consists of 202 images with a resolution of 1920×1080".
I suggest polishing this manuscript with a native English speaker.
3 How to obtain the Q^{m}_{n}？It is not explained clearly, which greatly affects subsequent understanding.
4. Figure 5 shows the result on the left and right, however, the caption explains "The top figure shows the results of the T-JND experiment, and the bottom figure shows the results of the S-JND experiment."
5. Figure 8 is not clear.

**Suitability:**

2

---

### Official Review · Reviewer_4xz1 · 2024-05-30

**Rating:** 4
**Confidence:** 4

**Summary:**

The paper proposes to detect JND for text screen content images.
Specifically, the paper starts to conventionally consider the problem. In my opinion the paper is too dramatic on this, as typically there is no real problem with captions and texts added in video content. Nevertheless the study of this issue is relevant.

I do not agree with the name given to the semantic test, as it is not a JND.
The study concludes that the JND test is much more demanding than the semantic based text. No novelty here.
The eye tracking experiment does not allow any type of conclusions and reporting cases is not very appropriate for a scientific conference.

**Strengths:**

If the database of the subjetive evaluation is made available the paper is useful.
Otherwise is meaningless.

**Limitations:**

Remarks
-	The paper requires proof reading.
-	The paper should use conventional writing rules like having a space after punctation.
-	The dataset Only contains images. Hence, why using VVC only? Considering this type. Of content an image coding standard should be used. I would advice JPEG XL.
-	How many subjects were outliers? As far as I could understand only the methodology was defined.

**Suitability:**

2

---

### Meta-Review · Area_Chair_36Gt · 2024-06-30

**Recommendation:** Accept (Poster)
**Confidence:** 4

**Metareview:**

This paper focuses on the development of a semantic aware Just Noticeable Difference (JND) model for text screen content images (TSCI) compressed by the versatile video coding (VVC) standard. While in the first review round the reviewers were relatively reluctant and gave some critical comments, the authors addressed most of these in the rebuttal, and the reviewers have adapted their scores accordingly. The potential and novelty of the approach are clear. Thus, given that the authors address all the risen comments for the camera ready, I recommend the paper for acceptance as a poster.